# Attracting and Dispersing: A Simple Approach for Source-free Domain Adaptation

**Shiqi Yang**[1], **Yaxing Wang**[2],[*] **Kai Wang**[1], **Shangling Jui**[3], **Joost van de Weijer**[1]
[1] Computer Vision Center, Universitat Autonoma de Barcelona, Barcelona, Spain
[2] Nankai University, Tianjin, China
[3] Huawei Kirin Solution, Shanghai, China
`{syang,kwang,joost}@cvc.uab.es,`
`yaxing@nankai.edu.cn, jui.shangling@huawei.com`

## Abstract

We propose a simple but effective source-free domain adaptation (SFDA) method. Treating SFDA as an unsupervised clustering problem and following the intuition that local neighbors in feature space should have more similar predictions than other features, we propose to optimize an objective of prediction consistency. This objective encourages local neighborhood features in feature space to have similar predictions while features farther away in feature space have dissimilar predictions, leading to efficient feature clustering and cluster assignment simultaneously. For efficient training, we seek to optimize an upper-bound of the objective resulting in two simple terms. Furthermore, we relate popular existing methods in domain adaptation, source-free domain adaptation and contrastive learning via the perspective of discriminability and diversity. The experimental results prove the superiority of our method, and our method can be adopted as a simple but strong baseline for future research in SFDA. Our method can be also adapted to source-free open-set and partial-set DA which further shows the generalization ability of our method. Code is available in `https://github.com/Albert0147/AaD_SFDA`.

## 1 Introduction

Supervised learning methods which are based on training with huge amounts of labeled data are advancing almost all fields of computer vision. However, the learned models typically perform decently on test data which have a similar distribution with the training set. Significant performance degradation will occur if directly applying those models to a new domain different from the training set, where the data distribution (such as variation of background, styles or camera parameter) is considerably different. This kind of distribution shift is formally denoted as domain/distribution shift. It limits the generalization of the model to unseen domains which is important in real-world applications. There are several research fields trying to tackle this problem. One of them is *Domain Adaptation* (DA), which aims to reduce the domain shift between the labeled source domain and unlabeled target domain. Typical works [12, 38] resort to learn domain-invariant features, thus improving generalization ability of the model between different domains. And in the past few years, the main research line of domain adaptation is either trying to minimize the distribution discrepancy between two domains [32, 33, 35], or deploying adversarial training on features to learn domain invariant representation [52, 68, 4, 36]. Some methods also tackle domain shift from the view of semi-supervised learning [67, 27] or clustering [7, 50, 5].

Many recent methods [24, 29, 64, 66, 55, 15, 61] focus on *source-free domain adaptation* (SFDA), where source data are unavailable during target adaptation, due to data privacy and intellectual

---
[*]Corresponding Author.

36th Conference on Neural Information Processing Systems (NeurIPS 2022).

Table 1: Detailed comparison of SFDA methods on **VisDA**. 'ODA/PDA' means whether the method reports the results for open-set or partial-set DA. $|\mathcal{L}|$ means number of training objective terms.

| Method | Extra Modules/Processing | ODA/PDA | $|\mathcal{L}|$ | Per-class |
|---|---|:---:|:---:|:---:|
| SHOT [26] | Access all target data for pseudo labeling | ✓ | 3 | 82.9 |
| 3C-GAN [24] | Data generation by conditional GAN | ✗ | 5 | 81.6 |
| $A^2$Net [61] | Self-supervised learning with extra classifiers | ✗ | 5 | 84.3 |
| G-SFDA [66] | Store features for nearest neighbor retrieval | ✗ | 2 | 85.4 |
| NRC [64] | Store features for 2-hop nearest neighbor retrieval | ✗ | 4 | 85.9 |
| HCL [15] | Store historical models | ✓ | 2 | 83.5 |
| **Ours** | Store features for nearest neighbor retrieval | ✓ | 2 | **88.0** |

property concerns of both users and businesses. Some SFDA methods resort to neighborhood clustering and pseudo labeling. However, pseudo labeling methods [29] may suffer from negative impact from noisy labels [28], and neighborhood clustering methods [66, 64] fail to investigate the potential information from dissimilar samples. Other methods either demand complex extra modules/processing [24, 61] or the storing of historical models for contrastive learning [15].

Based on the fact that target features from the source model already form some semantic structure and following the intuition that for a target feature from a (source-pretrained) model, similar features should have closer predictions than dissimilar ones, we propose a new objective dubbed as Attracting-and-Dispersing (**AaD**) to achieve it. we upperbound this objective, resulting in a simple final objective which only contains two types of terms, which encourage discriminability and diversity respectively. Further, we unify several popular domain adaptation, source-free domain adaptation and contrastive learning methods from the perspective of discriminability and diversity. Experimental results on several benchmarks prove the superiority of our proposed method. Our simple method improves the state-of-the-art on the challenging VisDA with 2.1% to 88.0%. Additionally, extra experiments on open-set and partial-set DA further prove the effectiveness of our method. A preliminary comparison between different SFDA method is shown in Tab. 1, which shows the simplicity and generalization ability of our method: it only requires the storing of features and a few nearest neighbors searches without any additional module like a generator [24] or a classifier [61].

We summary our contributions as follows:

- We propose to tackle source-free domain adaptation by optimizing an upperbound of the proposed clustering objective, which is surprisingly simple.

- We relate several popular existing methods in domain adaptation, source-free domain adaptation and contrastive learning via the perspective of discriminability and diversity, which is helpful to understand existing methods and beneficial for future improvement.

- The experimental results prove the efficacy of our method, especially we achieve new state-of-the-art on the challenging VisDA, and the method can be also extended to source-free open-set and partial-set domain adaptation.

## 2   Related Work

**Domain Adaptation.**   Early DA methods such as [33, 49, 53] adopt moment matching to align feature distributions. For adversarial learning methods, DANN [9] formulates domain adaptation as an adversarial two-player game. The adversarial training of CDAN [34] is conditioned on several sources of information. DIRT-T [47] performs domain adversarial training with an added term that penalizes violations of the cluster assumption. Additionally, [22, 36, 44] adopts prediction diversity between multiple learnable classifiers to achieve local or category-level feature alignment between source and target domains. SRDC [50] proposes to directly uncover the intrinsic target discrimination via discriminative clustering to achieve adaptation. CST [31] proposes a simple self-training strategy to improve the rough pseudo label under domain shift.

**Source-free Domain Adaptation.**   The above-mentioned normal domain adaptation methods need to access source domain data at all time during adaptation. In recent years plenty of methods emerge trying to tackle source-free domain adaptation. USFDA [20] and FS [21] resort to synthesize extra training samples in order to get compact decision boundaries, which is beneficial for both

---

**Algorithm 1** Attracting and Dispersing for SFDA

---

**Require:** Source-pretrained model and target data $\mathcal{D}_t$
 1: Build memory bank storing all *target* features and predictions
 2: **while** Adaptation **do**
 3:     Sample batch $\mathcal{T}$ from $\mathcal{D}_t$ and Update memory bank
 4:     For each feature $z_i$ in $\mathcal{T}$, retrieve $K$-nearest neighbors ($\mathcal{C}_i$) and their predictions from memory bank
 5:     Update model by minimizing Eq. 5
 6: **end while**

---

the detection of open classes and also target adaptation. SHOT [26] proposes to freeze the source classifier and it clusters target features by maximizing mutual information along with pseudo labeling for extra supervision. 3C-GAN [24] synthesizes labeled target-style training images. It is based on a conditional GAN to provide supervision for adaptation. BAIT [65] extends MCD [44] to source-free setting. $A^2$Net [61] proposes to learn an additional target-specific classifier for hard samples and adopts a contrastive category-wise matching module to cluster target features. HCL [15] adopts Instance Discrimination [60] for features from current and historical models to cluster features, along with a generated pseudo label conditioned on historical consistency. G-SFDA [66] and NRC [64] propose neighborhood clustering which enforces prediction consistency between local neighbors.

**Deep Clustering and Contrastive Learning.**   Recent Deep Clustering methods can be roughly divided into two groups, they the differ in how they learn the feature representation and cluster assignments, either simultaneously or alternatively. For example, DAC [2] and DCCM [58] alternately update cluster assignments and between-sample similarity. Simultaneous clustering methods IIC [18] and ISMAT [14] are based on mutual information maximizing between samples and theirs augmentations. LA [70] depends on a huge amount of nearest neighbor searches and multiple extra runs of *k-means* clustering to aggregate features. Recent unsupervised clustering works [25, 51, 46] start to rely on contrastive learning, where InfoNCE [37] is typically deployed. And recently NNCLR [8] proposes to use nearest neighbors in the latent space as positives in contrastive learning to cover more semantic variations than pre-defined transformations. However an inevitable problem of normal contrastive learning is class collision where negative samples are from the same class. To tackle this issue, recent works [23, 16] propose to estimate cluster prototypes and integrate them into contrastive learning.

## 3   Method

For source-free domain adaptation (SFDA), we are given source-pretrained model in the beginning and an unlabeled target domain with $N_t$ samples as $\mathcal{D}_t = \{x_i^t\}_{i=1}^{N_t}$. Target domain have same $C$ classes as source domain in this paper (known as the closed-set setting). The goal of SFDA is to adapt the model to target domain without source data. We divide the model into two parts: the feature extractor $f$, and the classifier $g$. The output of the feature extractor is denoted as feature ($z_i = f(x) \in \mathbb{R}^h$), where $h$ is dimension of the feature space. The output of classifier is denoted as ($p_i = \delta(g(z_i)) \in \mathbb{R}^C$ ) where $\delta$ is the softmax function. We denote $P \in \mathbb{R}^{bs \times C}$ as the prediction matrix in a mini-batch. Regarding the SFDA as an unsupervised clustering problem, we address SFDA problem by clustering target features based on the proposed AaD. In additionally, we relate our method with several existing DA, SFDA and contrastive learning methods.

### 3.1   Attracting and Dispersing for Source-free Domain Adaptation

Since the source-pretrained model already learns a good feature representation, it can provides a decent initialization for target adaptation. We propose to achieve SFDA by attracting predictions for features that are located close in feature space, while dispersing predictions of those features farther away in feature space.

We define $p_{ij}$ as the probability that the feature $z_i \in \mathbb{R}^h$ has similar (or the same) prediction to feature $z_j$: $p_{ij} = \frac{e^{p_i^T p_j}}{\sum_{k=1}^{N_t} e^{p_i^T p_k}}$. It can be interpreted as the possibility that $p_j$ is selected as the neighbor of $p_i$ in the output space [10].

We then define two sets for each feature $z_i$: close neighbor set $\mathcal{C}_i$ containing $K$-nearest neighbors of $z_i$ (with distances as cosine similarity), and background set $\mathcal{B}_i$ which contains the features that are not in $\mathcal{C}_i$ (features potentially from different classes). To retrieve nearest neighbors for training, we build two memory banks to store all *target* features along with their predictions just like former works [27, 66, 64, 42], which is efficient in both memory and computation, since only the features along with their predictions computed in each mini-batch are used to update the memory bank.

Intuitively, for each feature $z_i$, the features in $\mathcal{B}_i$ should have less similar predictions than those in $\mathcal{C}_i$[2]. To achieve this, we first define two likelihood functions:

$$P(\mathcal{C}_i|\theta) = \prod_{j \in \mathcal{C}_i} p_{ij} = \prod_{j \in \mathcal{C}_i} \frac{e^{p_i^T p_j}}{\sum_{k=1}^{N_t} e^{p_i^T p_k}}, \ P(\mathcal{B}_i|\theta) = \prod_{j \in \mathcal{B}_i} p_{ij} = \prod_{j \in \mathcal{B}_i} \frac{e^{p_i^T p_j}}{\sum_{k=1}^{N_t} e^{p_i^T p_k}} \tag{1}$$

where $\theta$ denotes parameters of the model, for readability we omit $\theta$ in following equations. The probability $p_j$ in Eq. 1 is the stored prediction for neighborhood feature $z_j$, which is retrieved from the memory bank.

We then propose to achieve target features clustering by minimizing the following negative log-likelihood, denoted as *AaD* (**A**ttracting-**a**nd-**D**ispersing):

$$\tilde{L}_i(\mathcal{C}_i, \mathcal{B}_i) = -\log \frac{P(\mathcal{C}_i)}{P(\mathcal{B}_i)} \tag{2}$$

Noting that, if we only have $P(\mathcal{C}_i)$, it will be similar to Instance Discrimination [60], but we also consider $P(\mathcal{B}_i)$ and we operate on predictions instead of features. If regarding weights of the classifier $g$ as classes prototypes, optimizing Eq. 2 is not only pulling features towards their closest neighbors and pushing them away from background features, but also towards (or away from) corresponding class prototypes. Therefore, we can achieve feature clustering and cluster assignment simultaneously.

To simplify the training, instead of manually and carefully sampling background features, we use all other features except $z_i$ in the mini-batch as $\mathcal{B}_i$, which can be regarded as an estimation of the distribution of the whole dataset. We can reasonably believe that overall similarity of features in $\mathcal{C}_i$ is potentially higher than that of $\mathcal{B}_i$, even if $\mathcal{B}_i$ has intersection with $\mathcal{C}_i$ since features in $\mathcal{C}_i$ are the closest ones to feature $z_i$. By optimizing Eq. 2, we are encouraging features in $\mathcal{C}_i$, which have a higher chance of belonging to the same class, to have more similar predictions to $z_i$ than those features in $\mathcal{B}_i$, which have a lower chance of belonging to the same class. Note all features will show up in both the first and second term; intra-cluster alignment and inter-cluster separability are expected to be achieved after training.

One problem optimizing Eq. 2 is that all target data are needed to compute Eq. 1, which is infeasible in real-world situation. Here we resort to get an upper-bound of Eq. 2:

$$\tilde{L}_i(\mathcal{C}_i, \mathcal{B}_i) = -\log \frac{P(\mathcal{C}_i)}{P(\mathcal{B}_i)} = -\sum_{j \in \mathcal{C}_i} [p_i^T p_j - \log(\sum_{k=1}^{N_t} e^{p_i^T p_k})] + \sum_{m \in \mathcal{B}_i} [p_i^T p_m - \log(\sum_{k=1}^{N_t} e^{p_i^T p_k})]$$
$$= -\sum_{j \in \mathcal{C}_i} p_i^T p_j + \sum_{m \in \mathcal{B}_i} p_i^T p_m + (N_{\mathcal{C}_i} - N_{\mathcal{B}_i}) \log(\sum_{k=1}^{N_t} e^{p_i^T p_k}) \tag{3}$$

Since we set $N_{\mathcal{C}_i} < N_{\mathcal{B}_i}$, with Jensen's inequality:

$$\tilde{L}_i(\mathcal{C}_i, \mathcal{B}_i) \leq -\sum_{j \in \mathcal{C}_i} p_i^T p_j + \sum_{m \in \mathcal{B}_i} p_i^T p_m + (N_{\mathcal{C}_i} - N_{\mathcal{B}_i})(\sum_{k=1}^{N_t} \frac{1}{N_t} p_i^T p_k + \log N_t)$$
$$\simeq \sum_{m \in \mathcal{B}_i} p_i^T p_m - \sum_{j \in \mathcal{C}_i} p_i^T p_j + (N_{\mathcal{C}_i} - N_{\mathcal{B}_i})(\sum_{k \in \mathcal{B}_i} \frac{p_i^T p_k}{N_{\mathcal{B}_i}} + \log N_t) \tag{4}$$
$$= -\sum_{j \in \mathcal{C}_i} p_i^T p_j + \frac{N_{\mathcal{C}_i}}{N_{\mathcal{B}_i}} \sum_{m \in \mathcal{B}_i} p_i^T p_m + (N_{\mathcal{C}_i} - N_{\mathcal{B}_i}) \log N_t$$

---

[2]For better understanding, we refer to $\mathcal{B}_i$ and $\mathcal{C}_i$ as index sets.

Table 2: Decomposition of methods into two terms: discriminability (*dis*) and diversity (*div*), which will be minimized for training.

| Method | Task | *dis* term | *div* term |
|--------|------|-----------|-----------|
| MI | SFDA&Clustering | $H(Y|X)$ | $-H(Y)$ |
| BNM | DA&SFDA | $-\|P\|_F$ | $-rank(P)$ |
| NC | SFDA | $-g(W_{ij}p_i^T p_j)$ | $\sum_{c=1}^{C} \mathrm{KL}(\bar{p}_c \| q_c)$ |
| InfoNCE | Contrastive | $-f(x)^T f(y)/\tau$ | $\log(\frac{e}{\tau} + \sum_i e^{f(x_i^-)^T f(x)/\tau})$ |
| **Ours** | SFDA | $-\sum_{j \in \mathcal{C}_i} p_i^T p_j$ | $\sum_{m \in \mathcal{B}_i} p_i^T p_m$ |

where $N_{\mathcal{C}_i}$ and $N_{\mathcal{B}_i}$ is the number of features in $\mathcal{C}_i$ and $\mathcal{B}_i$. Note that we cannot get this upper-bound without $P(\mathcal{B}_i)$. The approximation above in the penultimate line is to estimate the average dot product using the mini-batch data. This leads to the *surprisingly simple final objective* for unsupervised domain adaptation:

$$L = \mathbb{E}[L_i(\mathcal{C}_i, \mathcal{B}_i)], \text{with } L_i(\mathcal{C}_i, \mathcal{B}_i) = -\sum_{j \in \mathcal{C}_i} p_i^T p_j + \lambda \sum_{m \in \mathcal{B}_i} p_i^T p_m \quad (5)$$

Note the gradient will come from both $p_i$ and $p_m$. The first term aims to enforce prediction consistency between local neighbors, and the naive interpretation of second term is to disperse the prediction of potential dissimilar features, which are all other features in the mini-batch. Note that the dot product between two softmaxed predictions will be maximal when two predictions have the same predicted class and are close to one-hot vector. Our algorithm is illustrated in Algorithm. 1.

Unlike using a constant for the second term in Eq. 4 we empirically found that using a hyperparameter $\lambda$ to decay second term (starting from 1) works better, we will adopt **SND** [43] to tune this hyperparameter unsupervisedly. One reason may be that the approximation inside Eq. 3.1 is not necessarily accurate. And as training goes on, features are gradually clustering, the role of the second term for dispersing should be weakened. Additionally, considering the current mini-batch with the correctly predicted features $z_i$ and $z_m$ belonging to the same class. In this case the second term in both $L_i(\mathcal{C}_i, \mathcal{B}_i)$ and $L_m(\mathcal{C}_m, \mathcal{B}_m)$ tends to push $p_m$ to the wrong direction, while the first term in $L_m(\mathcal{C}_m, \mathcal{B}_m)$ can potentially keep current (correct) prediction unchanged. Hence, this will suppress the negative impact of the second term. We will further deepen the understanding of these two terms in the next subsection.

## 3.2 Relation to Existing Works

In this section, we will relate several popular DA, SFDA and contrastive learning methods through two objectives, *discriminability* and *diversity*. This can improve our understanding of domain adaptation methods, as well as improve the understanding of our method.

**Mutual Information maximizing (MI).** SHOT-IM [26] proposes to achieve source-free domain adaptation by maximizing the mutual information, which is actually widely used in unsupervised clustering [11, 40, 14]:

$$L_{MI} = H(Y|X) - H(Y) \quad (6)$$

which contains two terms: conditional entropy term $H(Y|X)$ to encourages unambiguous cluster assignments, and marginal entropy term $H(Y)$ to encourage cluster sizes to be uniform to avoid degeneracy. In practice, $H(Y)$ is approximated by the current mini-batch instead of using whole dataset [48, 14].

**Batch Nuclear-norm Maximization (BNM).** BNM [5, 6] aims to increase prediction discriminability and diversity to tackle domain shift. It is originally achieved by maximizing $F$-norm (for discriminability) and rank of prediction matrix (for diversity) respectively:

$$L = -\|P\|_F - rank(P) \quad (7)$$

In their paper, they further prove merely maximizing the nuclear norm $\|P\|_*$ can achieve these two goals simultaneously. In relation to our method, if target features are well clustering during training, we can presume the K-nearest neighbors of feature $z_i$ have the same prediction, the first term in Eq. 5 can be seen as the summation of diagonal elements of matrix $PP^T$, which is actually the square

of $F$-norm ($\|P\|_F = \sqrt{trace(PP^T)}$), then it is actually minimizing prediction entropy [5]. As for second term, we can regard it as the summation of non-diagonal element of $PP^T$, it encourages all these non-diagonal elements to be 0 thus the $rank(PP^T) = rank(P)$ is supposed to increase, which indicates larger prediction diversity [5]. In a nutshell, compared to SHOT and BNM our method first considers local feature structure to cluster target features, which can be treated as an alternative way to increase discriminability at the late training stage, meanwhile as discussed above our method is also encouraging diversity.

**Neighborhood Clustering (NC).** G-SFDA [66] and NRC [64] are based on neighborhood clustering to tackle SFDA problem. Those works basically contain two major terms in their optimizing objective: a neighborhood clustering term for prediction consistency and a marginal entropy term $H(Y)$ for prediction diversity. NRC [64] further introduces neighborhood reciprocity to weight the different neighbors. Their loss objective can be written as:

$$L_i = - \sum_{j \in \mathcal{C}_i} g(W_{ij} p_i^T p_j) + \sum_{c=1}^{C} \text{KL}(\bar{p}_c \| q_c), \text{ with } \bar{p}_c = \frac{1}{n_t} \sum_i p_i^{(c)} \text{, and } q_{\{c=1,..,C\}} = \frac{1}{C}$$

where $W_{ij}$ will weight the importance of neighbor and $g(\cdot)$ is *log* or *identity* function. Although the first term of G-SFDA and NRC is the same as that of our final loss objective Eq. 5, note that our motivation is different as we simultaneously consider similar and dissimilar features, and Eq. 5 is deduced as an approximated upper-bound of our original objective Eq. 2.

And note actually the marginal entropy term $-H(Y) = \sum_{c=1}^{C} \bar{p}_c \log \bar{p}_c = \sum_{c=1}^{C} \text{KL}(\bar{p}_c \| q_c) - \log C$. Although the second term of those methods are favoring prediction diversity to avoid the trivial solution where all images are only assigned to some certain classes, the margin entropy term presumes the prior that whole dataset or the mini-batch is class balance/uniformly distributed, which is barely true for current benchmarks or in real-world environment. In conclusion, the above three types of methods are actually all to increase discriminability and meanwhile maximize diversity of the prediction, but through different ways.

**Contrastive Learning.** Here we also link our method to InfoNCE [37]), which is widely used in contrastive learning. As a recent paper [56] points out that InfoNCE loss can be decomposed into 2 terms:

$$L_{infoNCE} = \mathbb{E}_{(x,y) \sim p_{pos}}[-f(x)^T f(y)/\tau] + \mathbb{E}_{\substack{x \sim p_{data} \\ \{x_i^-\}_{i=1}^M \sim p_{data}}} [\log(e^{1/\tau} + \sum_i e^{f(x_i^-)^T f(x)/\tau})] \tag{8}$$

The first term is denoted as *alignment* term (with positive pairs) is to make positive pairs of features similar, and the second term denoted as *uniformity* term with negative pairs encouraging all features to roughly uniformly distributed in the feature space.

The Eq. 8 shares some similarity with all the above domain adaptation methods in that the first term is for the alignment with positive pairs and the second term is to encourage diversity. But note that the remarkable difference is that the above domain adaptation methods operate in the output (prediction) space while contrastive learning is conducted in the (spherical) feature space. Therefore, simultaneously feature representation learning and cluster assignment can be achieved for those domain adaptation methods. Note in normal contrastive learning methods, extra KNN or a linear learnable classifier needs to be deployed for final classification, while our model can directly give predictions.

We list all above methods in Tab. 2. Finally, returning to Eq. 5, we can also regard the second term as a variant of diversity loss to avoid degeneration solution, but without making any category prior assumption. Intuitively, with target features forming groups during training, the second term should play less and less important role, otherwise it may destabilize the training. This is similar to the class collision issue in contrastive learning. If our second term contains too many features belonging to the same class. Thus it is reasonable to decay the second term.

## 4 Experiments

**Datasets.** We conduct experiments on three benchmark datasets for image classification: Office-31, Office-Home and VisDA-C 2017. **Office-31** [41] contains 3 domains (Amazon, Webcam, DSLR)

Table 3: Accuracies (%) on Office-Home for ResNet50-based methods. We highlight the best result and underline the second best one.

| Method | SF | Ar→Cl | Ar→Pr | Ar→Rw | Cl→Ar | Cl→Pr | Cl→Rw | Pr→Ar | Pr→Cl | Pr→Rw | Rw→Ar | Rw→Cl | Rw→Pr | Avg |
|---|---|---|---|---|---|---|---|---|---|---|---|---|---|---|
| ResNet-50 [13] | ✗ | 34.9 | 50.0 | 58.0 | 37.4 | 41.9 | 46.2 | 38.5 | 31.2 | 60.4 | 53.9 | 41.2 | 59.9 | 46.1 |
| MCD [44] | ✗ | 48.9 | 68.3 | 74.6 | 61.3 | 67.6 | 68.8 | 57.0 | 47.1 | 75.1 | 69.1 | 52.2 | 79.6 | 64.1 |
| CDAN [34] | ✗ | 50.7 | 70.6 | 76.0 | 57.6 | 70.0 | 70.0 | 57.4 | 50.9 | 77.3 | 70.9 | 56.7 | 81.6 | 65.8 |
| SAFN [63] | ✗ | 52.0 | 71.7 | 76.3 | 64.2 | 69.9 | 71.9 | 63.7 | 51.4 | 77.1 | 70.9 | 57.1 | 81.5 | 67.3 |
| MDD [69] | ✗ | 54.9 | 73.7 | 77.8 | 60.0 | 71.4 | 71.8 | 61.2 | 53.6 | 78.1 | 72.5 | 60.2 | 82.3 | 68.1 |
| TADA [57] | ✗ | 53.1 | 72.3 | 77.2 | 59.1 | 71.2 | 72.1 | 59.7 | 53.1 | 78.4 | 72.4 | 60.0 | 82.9 | 67.6 |
| SRDC [50] | ✗ | 52.3 | 76.3 | 81.0 | 69.5 | 76.2 | 78.0 | 68.7 | 53.8 | 81.7 | 76.3 | 57.1 | 85.0 | 71.3 |
| SHOT [26] | ✓ | 57.1 | 78.1 | 81.5 | 68.0 | 78.2 | 78.1 | 67.4 | 54.9 | 82.2 | 73.3 | 58.8 | 84.3 | 71.8 |
| $A^2$Net [61] | ✓ | 58.4 | 79.0 | 82.4 | 67.5 | 79.3 | 78.9 | 68.0 | 56.2 | 82.9 | 74.1 | 60.5 | 85.0 | **72.8** |
| G-SFDA [66] | ✓ | 57.9 | 78.6 | 81.0 | 66.7 | 77.2 | 77.2 | 65.6 | 56.0 | 82.2 | 72.0 | 57.8 | 83.4 | 71.3 |
| NRC [64] | ✓ | 57.7 | 80.3 | 82.0 | 68.1 | 79.8 | 78.6 | 65.3 | 56.4 | 83.0 | 71.0 | 58.6 | 85.6 | 72.2 |
| BNM-S [6] | ✓ | 57.4 | 77.8 | 81.7 | 67.8 | 77.6 | 79.3 | 67.6 | 55.7 | 82.2 | 73.5 | 59.5 | 84.7 | 72.1 |
| **Ours** | ✓ | 59.3 | 79.3 | 82.1 | 68.9 | 79.8 | 79.5 | 67.2 | 57.4 | 83.1 | 72.1 | 58.5 | 85.4 | 72.7 |

Table 4: Accuracies (%) on VisDA-C (Synthesis → Real) for ResNet101-based methods. We highlight the best result and underline the second best one.

| Method | SF | plane | bcycl | bus | car | horse | knife | mcycl | person | plant | sktbrd | train | truck | **Per-class** |
|---|---|---|---|---|---|---|---|---|---|---|---|---|---|---|
| ResNet-101 [13] | ✗ | 55.1 | 53.3 | 61.9 | 59.1 | 80.6 | 17.9 | 79.7 | 31.2 | 81.0 | 26.5 | 73.5 | 8.5 | 52.4 |
| CDAN+BSP [3] | ✗ | 92.4 | 61.0 | 81.0 | 57.5 | 89.0 | 80.6 | 90.1 | 77.0 | 84.2 | 77.9 | 82.1 | 38.4 | 75.9 |
| MCC [19] | ✗ | 88.7 | 80.3 | 80.5 | 71.5 | 90.1 | 93.2 | 85.0 | 71.6 | 89.4 | 73.8 | 85.0 | 36.9 | 78.8 |
| STAR [36] | ✗ | 95.0 | 84.0 | 84.6 | 73.0 | 91.6 | 91.8 | 85.9 | 78.4 | 94.4 | 84.7 | 87.0 | 42.2 | 82.7 |
| RWOT [62] | ✗ | 95.1 | 80.3 | 83.7 | 90.0 | 92.4 | 68.0 | 92.5 | 82.2 | 87.9 | 78.4 | 90.4 | 68.2 | 84.0 |
| 3C-GAN [24] | ✓ | 94.8 | 73.4 | 68.8 | 74.8 | 93.1 | 95.4 | 88.6 | 84.7 | 89.1 | 84.7 | 83.5 | 48.1 | 81.6 |
| SHOT [26] | ✓ | 94.3 | 88.5 | 80.1 | 57.3 | 93.1 | 94.9 | 80.7 | 80.3 | 91.5 | 89.1 | 86.3 | 58.2 | 82.9 |
| $A^2$Net [61] | ✓ | 94.0 | 87.8 | 85.6 | 66.8 | 93.7 | 95.1 | 85.8 | 81.2 | 91.6 | 88.2 | 86.5 | 56.0 | 84.3 |
| G-SFDA [66] | ✓ | 96.1 | 88.3 | 85.5 | 74.1 | 97.1 | 95.4 | 89.5 | 79.4 | 95.4 | 92.9 | 89.1 | 42.6 | 85.4 |
| NRC [64] | ✓ | 96.8 | 91.3 | 82.4 | 62.4 | 96.2 | 95.9 | 86.1 | 80.6 | 94.8 | 94.1 | 90.4 | 59.7 | 85.9 |
| HCL [15] | ✓ | 93.3 | 85.4 | 80.7 | 68.5 | 91.0 | 88.1 | 86.0 | 78.6 | 86.6 | 88.8 | 80.0 | 74.7 | 83.5 |
| **Ours** | ✓ | 97.4 | 90.5 | 80.8 | 76.2 | 97.3 | 96.1 | 89.8 | 82.9 | 95.5 | 93.0 | 92.0 | 64.7 | **88.0** |

with 31 classes and 4,652 images. **Office-Home** [54] contains 4 domains (Real, Clipart, Art, Product) with 65 classes and a total of 15,500 images. **VisDA** (VisDA-C 2017) [39] is a more challenging dataset, with 12-class synthetic-to-real object recognition tasks, its source domain contains of 152k synthetic images while the target domain has 55k real object images.

**Evaluation.** The column **SF** in the tables denotes source-free. For Office-31 and Office-Home, we show the results of each task and the average accuracy over all tasks (*Avg* in the tables). For VisDA, we show accuracy for all classes and average over those classes (*Per-class* in the table). All results are the average of three random runs for target adaptation.

**Model details.** To ensure fair comparison with related methods, we adopt the backbone of a ResNet-50 [13] for Office-Home and ResNet-101 for VisDA. Specifically, we use the same network architecture as SHOT [26], BNM-S [6], G-SFDA [66] and NRC [64], *i.e.*, the final part of the network is: *fully connected layer - Batch Normalization [17] - fully connected layer with weight normalization [45]*. We adopt SGD with momentum 0.9 and batch size of 64 for all datasets. The learning rate for Office-31 and Office-Home is set to 1e-3 for all layers, except for the last two newly added fc layers, where we apply 1e-2. Learning rates are set 10 times smaller for VisDA. We train 40 epochs for Office-31 and Office-Home while 15 epochs for VisDA.

There are two hyperparameters $N_{\mathcal{C}_i}$ (number of nearest neighbors) and $\lambda$, to ensure fair comparison we set $N_{\mathcal{C}_i}$ to the same number as previous works G-SFDA [66] and NRC [64], which also resort to nearest neighbors. That is, we set $N_{\mathcal{C}_i}$ to 3 on Office-31 and Office-Home, 5 on VisDA. For $\lambda$, we set it as $\lambda = (1 + 10 * \frac{iter}{max\_iter})^{-\beta}$, where the decay factor $\beta$ controls the decaying speed. *We directly apply **SND** [43] to select $\beta$ unsupervisedly.* Based on SND we set $\beta$ to 0 on Office-Home, 2 on Office-31 and 5 on VisDA.

### 4.1 Results and Analysis

**Quantitative Results.** As shown in Tables 3-5(*Left*), where the top part shows results for the source-present methods that use source data during adaptation, and the bottom part shows results for

Table 5: (**Left**) Accuracies (%) on Office-31 for ResNet50-based methods. We highlight the best result and underline the second best one. (**Right**) Ablation study on number of nearest neighbors $N_{\mathcal{C}_i}$. We highlight the best score and underline the second best one.

| Method | SF | A→D | A→W | D→W | W→D | D→A | W→A | Avg |
|---|---|---|---|---|---|---|---|---|
| MCD [44] | ✗ | 92.2 | 88.6 | 98.5 | 100.0 | 69.5 | 69.7 | 86.5 |
| CDAN [34] | ✗ | 92.9 | 94.1 | 98.6 | 100.0 | 71.0 | 69.3 | 87.7 |
| MDD [69] | ✗ | 90.4 | 90.4 | 98.7 | 99.9 | 75.0 | 73.7 | 88.0 |
| DMRL [59] | ✗ | 93.4 | 90.8 | 99.0 | 100.0 | 73.0 | 71.2 | 87.9 |
| MCC [19] | ✗ | 95.6 | 95.4 | 98.6 | 100.0 | 72.6 | 73.9 | 89.4 |
| SRDC [50] | ✗ | 95.8 | 95.7 | 99.2 | 100.0 | 76.7 | 77.1 | **90.8** |
| SHOT [26] | ✓ | 94.0 | 90.1 | 98.4 | 99.9 | 74.7 | 74.3 | 88.6 |
| 3C-GAN [24] | ✓ | 92.7 | 93.7 | 98.5 | 99.8 | 75.3 | 77.8 | 89.6 |
| NRC [64] | ✓ | 96.0 | 90.8 | 99.0 | 100.0 | 75.3 | 75.0 | 89.4 |
| HCL [15] | ✓ | 94.7 | 92.5 | 98.2 | 100.0 | 75.9 | 77.7 | 89.8 |
| BNM-S [6] | ✓ | 93.0 | 92.9 | 98.2 | 99.9 | 75.4 | 75.0 | 89.1 |
| **Ours** | ✓ | 96.4 | 92.1 | 99.1 | 100.0 | 75.0 | 76.5 | 89.9 |

| $N_{\mathcal{C}_i}$ | Avg |
|---|---|
| **Office-31** | |
| 1 | 89.1 |
| 2 | 89.5 |
| 3 | **89.9** |
| **Office-Home** | |
| 1 | 72.2 |
| 2 | 72.6 |
| 3 | **72.7** |

| $N_{\mathcal{C}_i}$ | Per-class |
|---|---|
| **VisDA** | |
| 3 | 86.7 |
| 4 | 87.4 |
| 5 | **88.0** |
| 6 | **88.0** |
| 7 | **88.0** |

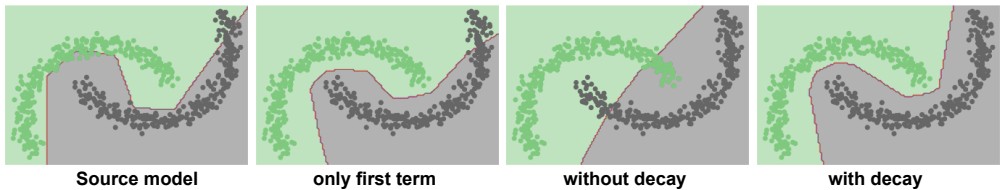

Figure 1: Visualization of decision boundary on target data with different training objective.

the source-free DA methods. On Office-31 and VisDA, our method gets state-of-the-art performance compared to existing source-free domain adaptation methods, especially on VisDA our method outperforms others by a large margin (2.1% compared to NRC). And our method achieves similar results on Office-Home compared to the more complex $A^2$Net method (*which combines three classifiers and five objective functions*). The reported results clearly demonstrate the efficiency of the proposed method for source-free domain adaptation. It also achieves similar or better results compared to domain adaptation methods with access to source data on both Office-Home and VisDA. Note the extension of SHOT called SHOT++ [30] deploys extra self-supervised training and semi-supervised learning, which are general to improve the results (*an evidence is that the source model after these 2 tricks gets huge improvement, e.g., 60.2% improves to 66.6% on Office-Home.*), we do not list it here for fair comparison.

**Toy dataset.** We carry out an experiment on the twinning moona dataset to ablate the influence of two terms in our objective Eq. 5. For the twinning moons dataset, the data from the source domain are represented by two inter-twinning moons, which contain 300 samples each. Data in the target domain are generated through rotating source data by $30°$. The domain shift here is instantiated as the rotation degree. First we train the model with 3 linear layers only on the source domain, and test the model on all domains. As shown in the first image in Fig. 1, the source model performs badly on target data. Then we conduct several variants of our method to train the model. The visualization of the decision boundary in Fig. 1 indicates that both terms in Eq. 5 are necessary, and decay of second term is shown to be important.

Table 6: **Unsupervised hyperparameter selection of $\beta$ with *SND* [43]**, larger *SND* should correspond to better target model.

| | Office-31 | | | Office-Home | | | | VisDA | |
|---|---|---|---|---|---|---|---|---|---|
| $\beta$ | *SND*↑ | Avg | $\beta$ | *SND*↑ | Avg | $\beta$ | *SND*↑ | Per-class | |
| 0 | 4.1366 | 88.0 | 0 | **3.7515** | **72.7** | 0 | 8.1823 | 77.5 | |
| 0.25 | 4.3016 | 89.7 | 0.25 | 3.7402 | 72.6 | 1 | 8.2584 | 83.8 | |
| 1 | 4.4494 | **89.9** | 0.5 | 3.7252 | 72.0 | 2 | 8.3214 | 86.7 | |
| 2 | **4.4501** | **89.9** | 1 | 3.6923 | 70.6 | 3 | 8.3311 | 87.6 | |
| | | | | | | 4 | 8.3540 | 88.0 | |
| | | | | | | 5 | **8.3543** | 88.0 | |
| | | | | | | 7 | 8.3530 | **88.1** | |

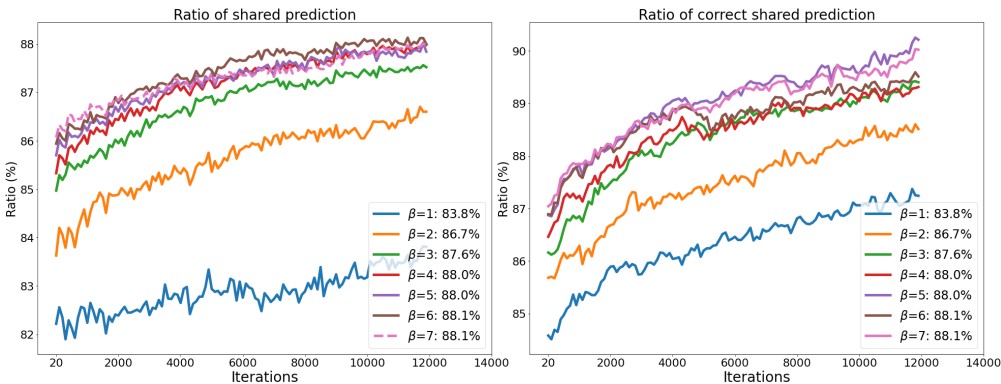

Figure 2: (**Left**) Ratio of features which have 3 nearest neighbor features sharing the same predicted label. (**Right**) Ratio among **above features** which have 3 nearest neighbor features sharing the same and **correct** predicted label.

Table 7: Runtime analysis on SHOT and our method. For SHOT, pseudo labels are computed at each epoch. 10% and 5% denote the percentage of target features which are stored in the memory bank.

| VisDA | Runtime (s/epoch) | Per-class (%) |
|---|---|---|
| SHOT | 618.82 | 82.9 |
| Ours | 520.13 | 88.0 |
| Ours(10% for memory bank) | 490.21 | 87.6 |
| Ours(5% for memory bank) | 482.77 | 87.5 |

**Number of nearest neighbors ($N_{\mathcal{C}_i}$).** For the number of nearest neighbors used for the first term in Eq. 5, we show in Tab. 5 (*Right*) our method is robust to the choice of $N_{\mathcal{C}_i}$, as the results imply that a reasonable choice of $N_{\mathcal{C}_i}$ (such as 3) works quite well on all datasets, since only considering few neighbors (such as 1/2) may be too noisy if all of them are misclassified, while setting $N_{\mathcal{C}_i}$ too larger may also potentially include samples of other categories. For larger dataset such as VisDA we can choose a relatively larger $N_{\mathcal{C}_i}$. Note the reason why we choose $N_{\mathcal{C}_i}$ as 5 in main experiments is to compare fairly with G-SFDA [66] and NRC [64].

**Decay factor $\beta$.** According to the analysis in Sec. 3.2, the second term acts like a diversity term to avoid that all target features collapse to a limited set of categories. The role of the second term should be weakened during the training, but how to decay the second term is non-trivial. We directly adopt *SND* [43] which computes Soft Neighborhood Density for unsupervised hyperparameter selection of $\beta$. The method is unsupervised and larger *SND* predicts a better target models. The results of *SND* with different $\beta$ are shown in Tab. 6, the results prove that *SND* works well to choose optimal $\beta$.

**Runtime analysis.** Instead of storing all features in the memory bank, we can only stores a limited number of target features, by updating the memory bank at the end of each iteration by taking the $n$ (batch size) embeddings from the current training iteration and concatenating them at the end of the memory bank, and discard the oldest $n$ elements from the memory bank. We report the results with this type of memory bank of different buffer size in the Table 7. The results show that indeed this could be an efficient way to reduce computation on very large datasets.

**Degree of clustering during training.** We also plot how features are clustered with different decaying factors $\beta$ on VisDA in Fig. 2. The left one shows the ratio of features which have 3-nearest neighbors all sharing the same prediction, which indicates the degree of clustering during training, and the right one shows the ratio among above features which have 3-nearest neighbor features sharing the same and *correct* predicted label. Those curves in Fig. 2 *left* show that the target features are clustering, and those in Fig. 2 *right* indicate that clear category boundaries are emerging. The numbers in the legends denote the deployed $\beta$ and the corresponding final accuracy. From the figures we can draw the conclusion that with a larger decay factor $\beta$ on VisDA, features are quickly clustering

Table 8: Accuracy on Office-Home using ResNet-50 as backbone for **Source-free open-set DA**. *OS\**, *UNK* and *HOS* mean average per-class accuracy across known classes, unknown accuracy and harmonic mean between known and unknown accuracy respectively.

| | Ar → Cl | | | Ar → Pr | | | Ar → Rw | | | Cl→ Ar | | | Cl → Pr | | | Cl → Rw | | |
|---|---|---|---|---|---|---|---|---|---|---|---|---|---|---|---|---|---|---|
| | OS* | UNK | HOS | OS* | UNK | HOS | OS* | UNK | HOS | OS* | UNK | HOS | OS* | UNK | HOS | OS* | UNK | HOS |
| SHOT | 67.0 | 28.0 | 39.5 | 81.8 | 26.3 | 39.8 | 87.5 | 32.1 | 47.0 | 66.8 | 46.2 | 54.6 | 77.5 | 27.2 | 40.2 | 80.0 | 25.9 | 39.1 |
| **AaD** | 50.7 | 66.4 | **57.6** | 64.6 | 69.4 | **66.9** | 73.1 | 66.9 | **69.9** | 48.2 | 81.1 | **60.5** | 59.5 | 63.5 | **61.4** | 67.4 | 68.3 | **67.8** |

| | Pr → Ar | | | Pr → Cl | | | Pr → Rw | | | Rw→ Ar | | | Rw → Cl | | | Rw → Pr | | | Avg. | | |
|---|---|---|---|---|---|---|---|---|---|---|---|---|---|---|---|---|---|---|---|---|---|
| | OS* | UNK | HOS | OS* | UNK | HOS | OS* | UNK | HOS | OS* | UNK | HOS | OS* | UNK | HOS | OS* | UNK | HOS | OS* | UNK | HOS |
| SHOT | 66.3 | 51.1 | 57.7 | 59.3 | 31.0 | 40.8 | 85.8 | 31.6 | 46.2 | 73.5 | 50.6 | 59.9 | 65.3 | 28.9 | 40.1 | 84.4 | 28.2 | 42.3 | 74.6 | 33.9 | 45.6 |
| **AaD** | 47.3 | 82.4 | **60.1** | 45.4 | 72.8 | **55.9** | 68.4 | 72.8 | **70.6** | 54.5 | 79.0 | **64.6** | 49.0 | 69.6 | **57.5** | 69.7 | 70.6 | **70.1** | 58.2 | 71.9 | **63.6** |

Table 9: Accuracy on Office-Home using ResNet-50 as backbone for **Source-free partial-set DA**.

| Partial-set DA | Ar→Cl | Ar→Pr | Ar→Re | Cl→Ar | Cl→Pr | Cl→Re | Pr→Ar | Pr→Cl | Pr→Re | Re→Ar | Re→Cl | Re→Pr | **Avg.** |
|---|---|---|---|---|---|---|---|---|---|---|---|---|---|
| SHOT-IM | 57.9 | 83.6 | 88.8 | 72.4 | 74.0 | 79.0 | 76.1 | 60.6 | 90.1 | 81.9 | **68.3** | **88.5** | 76.8 |
| SHOT | 64.8 | **85.2** | 92.7 | 76.3 | **77.6** | **88.8** | **79.7** | 64.3 | 89.5 | 80.6 | 66.4 | 85.8 | 79.3 |
| **AaD** | **67.0** | 83.5 | **93.1** | **80.5** | 76.0 | 87.6 | 78.1 | **65.6** | **90.2** | **83.5** | 64.3 | 87.3 | **79.7** |

and forming inter-class boundaries, since the ratio of features which share the same and correct prediction with neighbors are increasing faster. When decaying factor $\beta$ is too small, meaning training signal from the second term is strong, the clustering process is actually impeded. The curves in Fig. 2 (*left*) signify that this ratio can also be used to choose $\beta$ with higher performance unsupervisedly.

**Source-free partial-set and open-set DA.** We provide additional results under source-free partial-set and open-set DA (PDA and ODA) setting in Tab. 8 and Tab. 9 respectively, where the open-set detection in ODA follows the same protocol to detect unseen categories as SHOT. On ODA, instead of reporting average *per-class* accuracy $OS = \frac{|\mathcal{C}_s| \times OS^*}{|\mathcal{C}_s|+1} + \frac{1 \times UNK}{|\mathcal{C}_s|+1}$ where $|\mathcal{C}_s|$ is the number of known categories on source domain, we report results of $HOS = \frac{2 \times OS^* \times UNK}{OS^* + UNK}$, which is *harmonic mean* between known categories accuracy $OS^*$ and unknown accuracy *UNK*. As pointed out by [1], $OS$ is problematic since this metric can be quite high even when unknown class accuracy *UNK* is 0, while unknown category detection is the key part in open-set DA. We reproduce SHOT under open-set DA and report results of $OS^*$, *UNK* and *HOS* in Tab. 8, which shows our method gets much better balance between known and unknown accuracy.

## 5    Conclusion

We proposed to tackle source-free domain adaptation by encouraging similar features in feature space to have similar predictions while dispersing predictions of dissimilar features in feature space, to achieve simultaneously feature clustering and cluster assignment. We introduced an upper bound to our proposed objective, resulting in two simple terms. Further we showed that we can unify several popular domain adaptation, source-free domain adaptation and contrastive learning methods from the perspective of discriminability and diversity. The approach is simple but achieves state-of-the-art performance on several benchmarks, and can be also adapted to source-free open-set and partial-set domain adaptation.

## Acknowledgement

We acknowledge the support from Huawei Kirin Solution, and the project PID2019-104174GB-I00/AEI/10.13039/501100011033 (MINECO, Spain), and the CERCA Programme of Generalitat de Catalunya.

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
