# OpenReview forum: "Attracting and Dispersing: A Simple Approach for Source-free Domain Adaptation"
_NeurIPS.cc/2022/Conference — NeurIPS 2022 Accept_

### Official Review · Reviewer_G5ee · 2022-07-10

**Rating:** 6
**Confidence:** 4
**Soundness:** 3 good
**Presentation:** 4 excellent
**Contribution:** 3 good

**Summary:**

The authors propose a simple SFDA method that aims to adapt the model to the target data in a self-supervised manner. The adaptation is based on contrastive learning; positive pairs are constructed by searching kNN of the target sample from memory bank, while all sample pairs within the mini-batch is considered as negative pairs. Differently from popular contrastive learning methods, the output space is used to compute loss in the proposed method, which strongly encourages the model to assign each target data into predefined class labels. The experimental results validate the advantage of the proposed method.


**Questions:**

Related to the concerns described at <quality>:
- Do we really need P(B_i) to derive the loss function of the proposed method? What is the specific advange of considering P(B_i) here?
- How is the sensitivity of the proposed method to batch size?

Minor things:
- Is N_{C_i} different from K?

---

<After reading authors' rebuttal>

Since my concerns on <quality> have been resolved in the rebuttal, I updated my score from 4 to 6.


**Limitations:**

I think the proposed method does not perform well when batch size or the number of target data is too small.


**Strengths And Weaknesses:**

I will describe strengths and weaknesses from four perspectives: originality, quality, clarity and significance.

<originality>

Not highly novel. Methodologically, the proposed method is similar with contrastive learning with InfoNCE loss, but it is different in how to compute similarity scores in the loss function. Specifically, while the positive and negative pairs are constructed at the feature space in a similar way with [R1], the similarity scores in the loss function are computed at the output space. The authors provides some theretical justification on the formulation of the proposed method, though I have a concern on its derivation, which will be described in <quality>.
[R1] "Mean Shift for Self-Supervised Learning," ICCV 2021.

<quality>

Marginal. I have two concerns mainly on the derivation of the proposed method.
- In line 144-145, the authors state "we cannot get this upper-bound without P(B_i)," but I respectfully disagree with it. Using exactly the same deformations with Eq. (3) and (4), we can also have the same results by starting from -log P(C_i) except for the difference of the coefficients of the second and third terms at the third line of Eq. (4), which will be ignored in Eq. (5). This is due to the approximation adopted at the second line of Eq. (4), in which the sum over the target data is replaced with that within a mini-batch. Since the derivation of the proposed loss is quite similar with that of InfoNCE loss if we ignore P(B_i), showing the necessity of P(B_i) should be important to validate the novelty as well as the advantage of the proposed method.
- Due to the approximation mentioned above, it should be important to use a sufficiently large mini-batch as in contrastive learning with InfoNCE loss, which leads to a concern on sensitivity of the performance to batch size. However, the batch size seems to be fixed in all experiments.

<clarity>

Great. This paper is well-written and also well-organized. Someone might think that Section 2 and Section 3.2 should be integrated, but I think it depends on preference.

<significance>

Good. This paper provides some significant results in terms of showing that such a simple method can perform on par or better than other existing methods in several benchmark datasets. It is also a good news to be able to effectively tune the hyperparameter beta in an unsupervised way.

---

> ### Author Response · Authors · 2022-08-01
> **Reply to reviewer G5ee**
>
> Thanks for your comments, we will improve the paper to include all the discussions below.
>
> **(Novelty)** This paper features several novelties for the source-free domain adaptation task. (1) We optimize an upper-bound of the proposed clustering objective, which is surprisingly simple. It is not trivial to derive the final elegantly simple form (two dot products) from the original negative log likelihood function.  To the best of our knowledge, this direction has not been explored in the domain adaptation community. The derived equation for domain adaptation has some similarities with existing work (also aiming for discriminability and diversity) but the actual equation is different (and mostly more simple than existing ones as only containing dot product terms, see Table 2). (2) We relate several popular existing methods in domain adaptation, source-free domain adaptation and contrastive learning via the perspective of discriminability and diversity, which is helpful to understand existing methods and beneficial for future improvement. (3) While some components in our approach are familiar to the community (eg, the neighborhood clustering-based method and the contrastive learning method), it has a different (more simple) objective form; experiments show that this simple objective can obtain state-of-the-art on several datasets.
>
> **Q1.** We respectfully disagree with the claim that we do not need $P(\mathcal{B_i})$ to get the upper bound. Please note that one important step to get Eq. 4 from Eq. 3 is the Jensen’s inequality in line 143. If without $P(\mathcal{B_i})$ in last line of Eq. 3, the coefficient before $\log({{\sum_{k=1}^{N_t}}}e^{p_i^Tp_k})$ will be $N_{\mathcal{C_i}}$ instead of $N_{\mathcal{C_i}}-N_{\mathcal{B_i}}$. $N_{\mathcal{C_i}}$ is positive, thus $N_{\mathcal{C_i}}\log({{\sum_{k=1}^{N_t}}}e^{p_i^Tp_k})$ is **concave**, while $N_{\mathcal{C_i}}-N_{\mathcal{B_i}}$ is negative thus $(N_{\mathcal{C_i}}- N_{\mathcal{B_i}})\log({{\sum_{k=1}^{N_t}}}e^{p_i^Tp_k})$ is **convex**. Only with convex function we can use Jensen’s inequality to get an **upper bound**, otherwise for **concave** function we will get a **lower bound**. Intuitively in Eq. 2, considering $P(\mathcal{B_i})$ aims to explicitly disperse dissimilar features. Note in Eq. 1 and Eq. 2, the difference from infoNCE here is that we consider the whole dataset for clustering in the likelihood functions, but after the derivation in Eq. 3 and Eq. 4 we can get the simple upper bound.
>
>
> **Q2.** We show the results with different batch size on Ar$\rightarrow$Cl of Office-Home as below (we do not change the hyperparameters of our method here), where there are only a few images per class (around 3000 images with 65 classes). Due to limited gpu memory, we do not run experiments with larger batch size 128. The results below show that our method is less sensitive to batch size than SHOT-IM.
> As discussed in Sec. 3.2, all these methods have prediction diversity term, which could potentially deteriorate the clustering if used with a small batch size. For example, when the batch size is 16 and the number of classes is 65, encouraging the prediction matrix $P$ ([16, 65]) to have diverse category predictions may be problematic, because 16 samples can not have 65 class predictions. This is especially the case for SHOT-IM, where their diversity term (marginal entropy) encourages the marginal category distribution (average $P$ on the batch dimension, resulting in a vector with dimension [65]) to be a uniform distribution, since marginal entropy is actually a KL divergence using uniform distribution as prior (mentioned in line 197). While the diversity term in our method does not make any category prior assumption, it will be less sensitive than SHOT-IM, as the results below demonstrate.
>
> |   batch size | SHOT-IM| Ours|
> | ----   |  ----  |----  |
> |  64  | 57.1 |59.3	|
> |  32  | 55.2 |58.5	|
> |  16  | 42.2 |54.1	|
>
> **Q3.** $N_{C_i}$ is the same as K, we will improve the text to make it clear.
>
> Limitation: Source-free domain adaptation is not aiming to address the situation where target data is too small, and test-time adaptation is a possible solution for this.

---

> > ### Comment · Reviewer_G5ee · 2022-08-03
> > **Thanks for reply**
> >
> > Thanks for answering my questions and also for providing additional experimental results.
> > - As for Q1, there was my misunderstanding. I would like to really appreciate the authors' clarification to resolve it.
> > - As for Q2, the results provided in the reply are promising, which can further support the advantage of the proposed method.
> >
> > I updated my score from 4 to 6.

---

### Official Review · Reviewer_G5fg · 2022-07-11

**Rating:** 7
**Confidence:** 4
**Soundness:** 3 good
**Presentation:** 3 good
**Contribution:** 3 good

**Summary:**

This paper proposed an efficient and effective approach for source-free domain adaptation, which encourages nearest neighbors in the feature space to be similar, while the rest of the samples within the minibatch are dissimilar.

**Questions:**

1. This paper claims the search of nearest neighbors within the mini-batch is efficient but has not been validated. Presenting the complexity analysis and the compute time would help to validate this claim.

2. Some details of the model are not discussed in detail. For example, it is not clearly presented how to calculate the probability that the feature, $p_{ij}$, and how it has been involved in the back-propagation.

3. What's the memory requirement for building up two memory banks that store all the target features?

4. The experiment results on DA datasets show that the proposed method has higher average accuracy than the traditional DA approaches. How does the proposed method perform better than those algorithms without accessing the source samples and labels?

**Strengths And Weaknesses:**

A well-written paper that has clear math notations and solid derivations. And the results on both DA dataset and Source-free open-set DA datasets looks promising. However, although the intuition for the Attracting-and-Dispersing strategy is super clear but not much theoretical support is provided. Also, some details of the model are not discussed in detail and the claim on the efficiency needs to be validated.

---

> ### Author Response · Authors · 2022-08-01
> **Reply to reviewer G5fg**
>
> Thanks for your comments, we will improve the paper to include all the discussions below.
>
> **1. (Runtime analysis)** In every mini-batch, we do K-nearest neighbor retrieval for all target features in the mini-batch. Note the competitive SFDA methods either synthesize labeled images which is hard to train and quite time-consuming, or compute the pseudo label by using all target features every few iterations during adaptation (SHOT). We compare the runtime for one epoch of SHOT (including their proposed pseudo labeling) and our methods, specifically for SHOT the pseudo label is only computed once per epoch for all target samples, the results are below, where our method is faster than SHOT. Note we do not use any accelerated library for nearest neighbor retrieving, which can be utilized to further speed up the training.
> Here we use a simple way to decrease the storing memory. Instead of storing all feature vectors in the memory bank. The method only stores a fixed number of target features, we update the memory bank at the end of each iteration by taking the n (batch size) embeddings from the current training iteration and concatenating them at the end of the memory bank, and discard the oldest n elements from the memory bank. We report the results with this type of memory bank of different buffer size in the table below. The results show that indeed this could be an efficient way to reduce computation on very large datasets.
>
> |   VisDA |runtime (s/epoch with one TITAN Xp) |Per-class on VisDA |
> |  ----  |----  |----  |
> |SHOT|	618.82|	82.9|
> | Ours|  520.13   |    88.0   |
> | Ours (10%)|   490.21  |    87.6   |
> | Ours (5%)|  482.77   |      87.5 |
>
> **2.** As in line 111 we define how to get $p_{ij}$, for nearest neighbor retrieving we adopt cosine similarity as mentioned in line 114. The gradient update is mentioned in line 148, that gradient will only come from the $p_i$ and $p_m$.
>
> **3.** Note the stored features (dimension: 256) and predictions (dimension: number of classes) are just vectors, the memory size of all is small. For example on VisDA which has 55388 images in target domain, the feature bank storing all will consume 56.7 MB memory and prediction bank with 2.7 MB. While as shown in the previous runtime analysis we can just store 5% of the data, leading to much less memory consumption (2.8 MB for feature bank and 133 KB for prediction banks) with similar performance. We will add those illustrations into the next version.
>
> **4.** It is a good question why source-free DA methods can surpass normal DA methods. The possible reason is that most SFDA methods like SHOT and ours are trying best to fully exploit the information of only the target domain. When applied with source data, the information from the source domain will impede the target adaptation of the SFDA method due to the domain shift. As for normal DA methods, since most of them are trying to alleviate the domain shift, the source data are necessary. While for SFDA methods, we do not require the network to be able to process both source and target domain data (this requirement might impede it from adapting optimally to the target domain). Instead, we only care about the target performance.

---

> > ### Comment · Reviewer_G5fg · 2022-08-08
> > **Some follow up questions**
> >
> > Thank you for your response. I have some follow-up questions:
> >
> > 1. I want to clarify question #2. For the computation of $p_{ij}$, what are $p_i$ and $p_j$, they were not defined and how are they calculated from the inputs?
> >
> > 2. Can you please provide more intuition for the Attracting-and-Dispersing strategy? And perhaps some theoretical analysis?

---

> > > ### Author Response · Authors · 2022-08-08
> > > **Reply to follow up questions**
> > >
> > > Thanks for the reply, here are the answers to the questions:
> > >
> > > 1. $p_i$ is the softmaxed output of classifier (prediction), we will clarify it in the paper.
> > >
> > > 2. After source training, we assume that the target features from source model already form semantic structure in some degree (as in line 38, since source model already has not bat performance on the target), we posit that we just need to do clustering to achieve SFDA. The motivation of our method is to encourage the closer features in the feature space to have similar prediction (***attract***), while to enforce those dissimilar features to have different predictions (***dispersing***). For efficient training, we use nearest neighborhood features as closer features, while other features in current mini-batch are treated as dissimilar features (as they can not be more similar than nearest neighbor features, even they contain these neighbors). The prediction similarity is formulated as Eq. 1 and Eq. 2 in the manner of likelihood function. The final optimized objective is formulated as a negative log likelihood (Eq.3). Through Eq. 4 ~ Eq. 5 we can get a simple upper bound.

---

> > > > ### Comment · Reviewer_G5fg · 2022-08-09
> > > > **Thanks for reply**
> > > >
> > > > Thank you for your reply, I updated my score from 6 to 7.

---

### Official Review · Reviewer_KU52 · 2022-07-11

**Rating:** 6
**Confidence:** 4
**Soundness:** 3 good
**Presentation:** 3 good
**Contribution:** 2 fair

**Summary:**

This paper approaches the source-free domain adaptation problem through the lens of clustering, which forces similar features to cluster and dissimilar features to disperse. A simple upper bound for the objective is introduced and optimized. On VisDA and Office-31, the proposed method achieves state-of-the-art performance.

**Questions:**

- The legends in Figure 2 are too small to read.

**Limitations:**

I couldn't find any discussion of the work's limitations or potential negative societal impact by the authors.

**Strengths And Weaknesses:**

### Strengths
- The paper is well written and easy to read.
- Impressive performance improvement in VisDA over previous SOTA.

### Weaknesses
- The proposed method's concept and effect are very similar to previous contrastive learning methods, such as InfoNCE. InfoNCE or SimCLR also set positive/negative pairs and force the positive pairs to attract while the negative pairs disperse. The difference between SimCLR and the proposed method appears to be marginal; SimCLR applies the loss to the projection layer, whereas the proposed method applies the loss to the output layer.
- Using the nearest neighbor as a positive pair in contrastive learning is not a new discovery for the deep learning community. [1]
- The experimental results are not promising. The proposed method performs well on VisDA, but it only outperforms HCL by 0.1 percent on Office-31 and fails to achieve state-of-the-art performance on Office-Home.
- Storing all target features in a memory bank and retrieving K-nearest neighbors from the memory bank for each batch appears to be a significant computational burden. Have the authors thought about this and computed the computational complexity of the proposed algorithm? Is this method capable of handling large target datasets? In comparison to existing methods, how much time or FLOPs does the proposed method take?

### Overall
The proposed loss objective is straightforward, and the concept is simple to grasp. The theoretical novelty and significance of the contributions, however, appear to be insufficient for this venue.

[1] Dwibedi, Debidatta, et al. "With a little help from my friends: Nearest-neighbor contrastive learning of visual representations." Proceedings of the IEEE/CVF International Conference on Computer Vision. 2021.

---

> ### Author Response · Authors · 2022-08-01
> **Reply to reviewer KU52**
>
> Thanks for your comments, we will improve the paper to include all the discussions below.
>
> **1&2.** **We agree with the reviewer that there are many papers in different communities which use contrastive learning where positive pairs attract each other and negative pairs are pushed apart. Indeed, contrastive losses have a long history and go back to early machine learning (90s). We do not consider that to be the novelty of our paper to the domain adaptation community.** We here briefly summarize the novelty:
> This paper features several novelties for the source-free domain adaptation task. (1) We optimize an upper-bound of the proposed clustering objective, which is surprisingly simple. It is not trivial to derive the final elegantly simple form (two dot products) from the original negative log likelihood function.  To the best of our knowledge, this direction has not been explored in the **domain adaptation community**. The derived equation for domain adaptation has some similarities with existing work (also aiming for discriminability and diversity) but the actual equation is different (and mostly more simple than existing ones as only containing dot product terms, see Table 2). (2) We relate several popular existing methods in domain adaptation, source-free domain adaptation and contrastive learning via the perspective of discriminability and diversity, which is helpful to understand existing methods and beneficial for future improvement. (3) While some components in our approach are familiar to the community (eg, the neighborhood clustering-based method and the contrastive learning method), it has a different (more simple) objective form; experiments show that this simple objective can obtain state-of-the-art on several datasets
> Thanks for the reference [1], we will add it in the related works. Note that we do not claim that using nearest neighbors for contrastive learning is a contribution.
>
>
> **3.**  ***(Performance)*** Our method is **simpler** compared to other SFDA methods, such as SHOT (see Runtime analysis below), 3C-GAN which needs to synthesize labeled target style images, HCL which needs to store several old models for contrastive learning and also pseudo labeling, and $A^2$Net which is based on the complex framework. Note that our method can be also deployed to open-set DA in Tab. 7, achieving 18\% higher H value than SHOT which is a very significant improvement. Our paper provides a new view to tackle the source-free domain adaptation and related several methods in domain adaptation and contrastive learning, before this paper no one went deep into this direction.
>
> **4.** ***(Runtime analysis)*** In every mini-batch, we do K-nearest neighbor retrieval for all target features in the mini-batch. Note the competitive SFDA methods either synthesize labeled images which is hard to train and quite time-consuming, or compute the pseudo label by using all target features every few iterations during adaptation (SHOT). We compare the runtime for one epoch of SHOT (including their proposed pseudo labeling) and our methods, specifically for SHOT the pseudo label is only computed once per epoch for all target samples, the results are below, where our method is faster than SHOT. Note we do not use any accelerated library for nearest neighbor retrieving, which can be utilized to further speed up the training.
> Here we use a simple way to decrease the storing memory. Instead of storing all feature vectors in the memory bank. The method only stores a fixed number of target features, we update the memory bank at the end of each iteration by taking the n (batch size) embeddings from the current training iteration and concatenating them at the end of the memory bank, and discard the oldest n elements from the memory bank. We report the results with this type of memory bank of different buffer size in the table below. The results show that indeed this could be an efficient way to reduce computation on very large datasets.
>
> |   VisDA |runtime (s/epoch with one TITAN Xp) |Per-class on VisDA |
> |  ----  |----  |----  |
> |SHOT|	618.82|	82.9|
> | Ours|  520.13   |    88.0   |
> | Ours (10%)|   490.21  |    87.6   |
> | Ours (5%)|  482.77   |      87.5 |

---

> > ### Comment · Reviewer_KU52 · 2022-08-03
> > **Thanks for reply**
> >
> > Thank you for responding to my concerns about this paper.
> >
> > (Performance) The proposed method's performance is not significant when compared to previous methods, but as the authors mentioned in their response, the proposed method has advantages in terms of simplicity and runtime.
> >
> > (Novelty) The intuition and derivation of the final simple objective are valuable, but the final form appears to be quite similar to the reference [1], with the only differences being whether to apply the loss on the output layer or feature and softmax with temperature parameter. If the proposed form is better than the proposed form in [1], theoretical or experimental grounds should be provided.
> >
> > I raised my rating from 3 to 4.

---

> > > ### Author Response · Authors · 2022-08-03
> > > **Further reply to Reviewer KU52**
> > >
> > > Thanks for your advice. We just try directly using NNCLR (Eq. 3 in [1]) to SFDA, but on the output space. We show the results with using only 1 nearest neighbor as in [1], and also 5 nearest neighbors (thus Eq. 3 in [1] will be computed 5 times, we found this lead to high performance than directly using 5-NN inside $\text{NN}(z_i, Q)$ in Eq.3 ) as in our paper, as in the table below. And note in our Eq. 1 and Eq. 2, the difference from [1] is that we consider the whole dataset for clustering in the likelihood functions, but after the derivation in Eq. 3 and Eq. 4 we can get the simple upper bound.
> > >
> > >
> > > |   Methods | Per-class on VisDA |
> > > |  ----  |----  |
> > > | [1], 1-NN  | 75.2 |
> > > | [1], 5-NN  | 76.4 |
> > > | AaD, 5-NN  | **88.0** |
> > >
> > > Additionally, we show the derivation of Eq. 3 in [1] as following (note actually here $n=N_{\mathcal{B}_i}+1$), where $\tau=0.1$ as the same in [1]:
> > >
> > > $$
> > > \mathcal{L}_{i}^\text{NNCLR} = -  \log{\frac{\exp{(\text{NN}(z_i, Q)^T z_i^+ /\tau)}}{\sum_k^{n}  \exp{(\text{NN}(z_i, Q)^T z_k^+ / \tau)}}} = - \text{NN}(z_i, Q)^T z_i^+/\tau + \log(\sum_k^{n} \exp{(\text{NN}(z_i, Q)^T z_k^+ / \tau)}) \geq - \text{NN}(z_i, Q)^T z_i^+/\tau+ \log n + \frac{\sum_k^n \text{NN}(z_i, Q)^T z_k^+ / \tau}{n}
> > > $$
> > >
> > > note the rightmost above can be further written as $- \frac{n-1}{n}(\text{NN}(z_i, Q)^T z_i^+/\tau)+ \log n +\frac{\sum^n_{k\neq i} \text{NN}(z_i, Q)^T z_k^+ / \tau}{n}$ where the first term only contains positive sample (thus $\textit{discriminative}$ term) and the last term only contains negative samples (thus $\textit{diversity}$ term). As you can see, the mathematical form is totally different (the coefficients), note that the ***lower bound*** of Eq. 3 in [1] (the above derivation) can not be utilized for optimizing since we need to ***minimize*** it.

---

> > > > ### Comment · Reviewer_KU52 · 2022-08-04
> > > > **Thanks for reply**
> > > >
> > > > Thank you for devoting your time and effort to address my concern about this paper. I updated the rating accordingly.

---

### Official Review · Reviewer_FqZ2 · 2022-07-11

**Rating:** 6
**Confidence:** 3
**Soundness:** 3 good
**Presentation:** 4 excellent
**Contribution:** 3 good

**Summary:**

This paper proposed a clustering-based method for the Source-Free Unsupervised Domain Adaptation (SFDA) problem. Considering the semantic information embedded in the feature space of the source model, the authors proposed an AaD clustering objective, aiming to simultaneously encourage the model’s discriminability and diversity during the adaptation. Besides, this paper also compared and analogized their method with several previous SFDA methods and gave some insightful analysis. Finally, experimental results on several domain adaptation benchmarks under different SFDA settings verified the efficacy of the proposed method.

**Questions:**

1. For the description of Algorithm 1 in line 5, it is written, “Update model by minimizing Eq.5”. Does this mean only updating the feature extractor f as implemented in [1], or updating the feature extractor and the classifier g simultaneously?
2. The choice and the analysis of $\lambda$: according to the simplified upper-bound of the proposed AaD objective, theoretically, $\lambda$ = $\frac{N_{C_i}}{N_{B_i}}$. Although the authors have explained the treatment of $\lambda$ and the introduction of decay exponent $\beta$, I am still curious about the numerical connection between the truly utilized $\frac{N_{C_i}}{N_{B_i}}$ (the theoretical value) and the evolving $\lambda$.
3. As we know, pseudo labeling is another widely leveraged technique for the SFDA problem, and it is usually combined with the hard or soft neighborhood clustering methods. I am curious whether the authors have considered combining pseudo labeling methods with the proposed AaD approach.

Other comments:

Typos:
In line 298, “Soruce-free” → “Source-free”.

**Limitations:**

The authors discussed the limitations in section 2.

**Strengths And Weaknesses:**

Strengths

1. Overall, this paper is well-written and easy to follow.
2. The proposed method is simple but effective, and the analysis between the proposed AaD objective and the previous SFDA methods in section 3.2 is insightful.
3. The experiment part is rich, which covers a wide range of SFDA SOTA methods and different SFDA settings (e.g., close-set, open-set, and partial set). The experimental results can generally confirm the effectiveness of the proposed method.

Weakness

1. Novelty: Compared with the previous neighborhood clustering-based method and the contrastive learning method, the proposed approach seems to be an incremental work, and the novelty is relatively limited.
2. The description of the algorithm and the training process is not completely clear. Please refer to the Questions section.

---

> ### Author Response · Authors · 2022-08-01
> **Reply to Reviewer FqZ2**
>
> Thanks for your comments, we will improve the paper to include all the discussions below.
>
> **W1.**
> This paper features several novelties for the source-free domain adaptation task. (1) We optimize an upper-bound of the proposed clustering objective, which is surprisingly simple. It is not trivial to derive the final elegantly simple form (two dot products) from the original negative log likelihood function.  To the best of our knowledge, this direction has not been explored in the domain adaptation community. The derived equation for domain adaptation has some similarities with existing work (also aiming for discriminability and diversity) but the actual equation is different (and mostly more simple than existing ones as only containing dot product terms, see Table 2). (2) We relate several popular existing methods in domain adaptation, source-free domain adaptation and contrastive learning via the perspective of discriminability and diversity, which is helpful to understand existing methods and beneficial for future improvement. (3) While some components in our approach are familiar to the community (eg, the neighborhood clustering-based method and the contrastive learning method), it has a different (more simple) objective form; experiments show that this simple objective can obtain state-of-the-art on several datasets.
>
> **Q1.** We train both classifier and feature extractor. We will explicitly mention this in any future version of the paper.
>
> **Q2.**
> Intuitively, at the beginning of adaptation, clustering is more needed, thus the second term $\lambda \sum_{m\in \mathcal{B}_i}p_i^Tp_m$, which aims to avoid model degeneration, could have a large weight to facilitate clustering. Then with target features forming groups during training, the second term should play a less and less important role, otherwise it may destabilize feature clustering. Then it is reasonable to decay the second term. Thus, unlike using a constant for the second term in Eq. 4 we empirically proved that using a hyperparameter λ to decay the second term indeed works better.
> As shown in the table below, on VisDA we still achieve competitive results with the theoretical one ($\frac{N_C}{N_B}$).
>
> |  $\lambda$  | Per-class on VisDA |
> |  ----  |----  |
> | $\frac{N_C}{N_B}=0.079$  | 86.1 |
> | $decay$ | **88.0** |
>
>
> **Q3.** The neighborhood clustering based SFDA method (like ours) is actually also a kind of soft pseudo labeling method, as they use the predictions of nearest neighbors as soft pseudo labels. Here we also consider combining soft neighborhood pseudo labeling (**local**) and hard pseudo labeling from SHOT (**global**). The results of combining AaD with pseudo labeling from SHOT (SHOT-PL) on VisDA are shown below. 1) First our AaD can improve the performance of SHOT-PL from 80.7\% (SHOT-PL) to 85.6\% (AaD+SHOT-PL). 2), while it is inferior to directly using AaD.
>
> The reason that combining AaD with SHOT pseudo labeling leads to lower results compared to just AaD, may be that AaD is only exploring local neighbor structure while SHOT-PL explores global cluster structure to compute pseudo labels. Sometimes the supervision from these two objectives may be contradictory ($\textit{e.g.}$, when features are near the decision boundary).
>
> It is not trivial to simultaneously utilize global (prototype based pseudo labeling as SHOT) and local structure (neighborhood based pseudo labeling as ours), which is indeed a potential solution for SFDA. We aim to further investigate this in future work.
>
> |    | Per-class on VisDA |
> |  ----  |----  |
> | AaD  | **88.0** |
> | AaD+SHOT-PL | 85.6 |
> | SHOT-PL | 80.7 |
> | SHOT | 82.9 |

---

> > ### Comment · Reviewer_FqZ2 · 2022-08-08
> > **Thanks for reply**
> >
> > Thanks for the authors' responses to my concerns about this paper.
> >
> > The authors explained some experimental details and addressed my concern about the theoretical and empirical gap of the hyperparameter lambda. I will keep my original positive rating score for this paper.

---

### Meta-Review · Area_Chair_3CqK · 2022-08-27

**Recommendation:** Accept
**Confidence:** Certain

**Metareview:**

This paper proposes a source-free domain adaptation method based on unsupervised clustering. The main assumption is that the source-trained model could generate target domain features that have smooth predictions in a neighbourhood. The proposed method optimizes the upper bound of the objective of prediction consistency. Experimental results show that the proposed method outperforms pseudo label and neighbourhood clustering methods.

While the main idea is not significantly novel, the effectiveness of the proposed algorithm is demonstrated by solid experimental studies. This is again a simple and efficient deep learning method designed with intuitions and without strong theoretical evidence. I would recommend acceptance of this paper given its impressive performance and solidness.


**Award:**

No

---

### Decision · Program_Chairs · 2022-09-14

Accept